# N-Acetylcysteine Protects Bladder Epithelial Cells from Bacterial Invasion and Displays Antibiofilm Activity against Urinary Tract Bacterial Pathogens

**DOI:** 10.3390/antibiotics10080900

**Published:** 2021-07-23

**Authors:** Arthika Manoharan, Samantha Ognenovska, Denis Paino, Greg Whiteley, Trevor Glasbey, Frederik H. Kriel, Jessica Farrell, Kate H. Moore, Jim Manos, Theerthankar Das

**Affiliations:** 1Department of Infectious Diseases and Immunology, School of Medical Sciences, Charles Perkins Centre, The University of Sydney, Sydney, NSW 2006, Australia; manoharan.arthika@sydney.edu.au (A.M.); denis.paino@health.nsw.gov.au (D.P.); jessica.farrell@sydney.edu.au (J.F.); jim.manos@sydney.edu.au (J.M.); 2Department of Urogynaecology, St George Hospital, University of New South Wales, Sydney, NSW 2052, Australia; sam.ognenovska@gmail.com (S.O.); k.moore@unsw.edu.au (K.H.M.); 3Whiteley Corporation, 19–23 Laverick Avenue, Tomago, NSW 2319, Australia; greg.whiteley@whiteley.com.au (G.W.); trevor.glasbey@whiteley.com.au (T.G.); erik.kriel@whiteley.com.au (F.H.K.)

**Keywords:** NAC, *E. coli*, UPEC, *E. faecalis*, biofilms, antibiotic resistance, IBCs

## Abstract

**Introduction:** Urinary tract infections (UTIs) affect more than 150 million individuals annually. A strong correlation exists between bladder epithelia invasion by uropathogenic bacteria and patients with recurrent UTIs. Intracellular bacteria often recolonise epithelial cells post-antibiotic treatment. We investigated whether N-acetylcysteine (NAC) could prevent uropathogenic *E. coli* and *E. faecalis* bladder cell invasion, in addition to its effect on uropathogens when used alone or in combination with ciprofloxacin. **Methods:** An invasion assay was performed in which bacteria were added to bladder epithelial cells (BECs) in presence of NAC and invasion was allowed to occur. Cells were washed with gentamicin, lysed, and plated for enumeration of the intracellular bacterial load. Cytotoxicity was evaluated by exposing BECs to various concentrations of NAC and quantifying the metabolic activity using resazurin at different exposure times. The effect of NAC on the preformed biofilms was also investigated by treating 48 h biofilms for 24 h and enumerating colony counts. Bacteria were stained with propidium iodide (PI) to measure membrane damage. **Results:** NAC completely inhibited BEC invasion by multiple *E. coli* and *E. faecalis* clinical strains in a dose-dependent manner (*p* < 0.01). This was also evident when bacterial invasion was visualised using GFP-tagged *E. coli*. NAC displayed no cytotoxicity against BECs despite its intrinsic acidity (pH ~2.6), with >90% cellular viability 48 h post-exposure. NAC also prevented biofilm formation by *E. coli* and *E. faecalis* and significantly reduced bacterial loads in 48 h biofilms when combined with ciprofloxacin. NAC visibly damaged *E. coli* and *E. faecalis* bacterial membranes, with a threefold increase in propidium iodide-stained cells following treatment (*p* < 0.05). **Conclusions:** NAC is a non-toxic, antibiofilm agent in vitro and can prevent cell invasion and IBC formation by uropathogens, thus providing a potentially novel and efficacious treatment for UTIs. When combined with an antibiotic, it may disrupt bacterial biofilms and eliminate residual bacteria.

## 1. Introduction

Urinary tract infections (UTIs) are one of the most commonly acquired bacterial infections worldwide, with an estimated 150 million infections annually [1]. In the USA alone, 1.5–2 million annual visits to the emergency are caused by UTIs [2], costing nearly $3.5 billion to the US economy [3]. In Australia, UTIs accounted for 12% of all potentially preventable hospitalisations in 2017–2018 alone, being the second most common cause of hospitalization [4]. Almost 30% of all women will have a recurrent episode of UTI. Most UTIs manifest in the form of an uncomplicated bladder infection, called cystitis. Bacteria can often ascend through the urethra and infect the bladder causing cystitis. If left untreated, it can ascend further and infect the kidneys and cause pyelonephritis, also known as a complicated UTI [5]. 

Uropathogenic *Escherichia coli* (UPEC) is the most prevalent UTI-causing bacterial species, accounting for over 90% of diagnosed community-acquired and 50% of nosocomial UTIs, respectively [6,7]. These mainly arise from the colonisation of the periurethral mucosa by faecal bacteria that ascend through the urinary tract [8,9]. UPEC accounts for 80% and 33% of all uncomplicated and complicated infections, respectively [10,11]. Other uropathogens such as *Enterococcus* spp. account for around 15% of all complicated UTIs in the hospital setting [12]. 

UPEC are specially equipped to survive the harsh bladder environment, as they possess an array of virulence factors, including outer-membrane vesicles, pili, curli, non-pilus adhesins, and outer-membrane proteins (OMPs), which assist in epithelial invasion and biofilm formation in the bladder [13]. UPEC has been shown to have a better adhesion capacity and display increased biofilm formation compared with non-UTI isolates [14]. This is achieved through its curli pili, which are amyloid fibres that promote adhesion, aggregation, and biofilm formation to the uroepithelium [14]. 

Amongst the Enterococcus spp., *Enterococcus faecalis* is a uropathogen of concern, especially given its biofilm-forming capacity and resistance to a wide range of antibiotics. Normally found as part of the commensal gut flora, *E. faecalis* is also known to invade urothelial cells during an active infection; however, whether it forms intracellular bacterial communities (IBCs) (which are biofilm-like communities) or not remains unknown [6]. *E. faecalis* is thought to establish UTIs in very different ways compared with UPEC, with more cystitis-causing isolates displaying a very high tropism for the kidney, leading to complicated UTIs [15]. *Enterococcus* spp. in general also employ special adhesion factors, such as the enterococcal surface protein (Esp) and the enterococcal polysaccharide antigen (Epa) to promote adhesion in the urinary tract [16].

Studies have shown that over 60% of clinical UTI isolates taken from patients with catheter-associated UTIs are positive for biofilm formation [17]. This is because biofilms are crucial for the persistence of UPEC, *E. faecalis*, and other pathogens such as *Proteus mirabilis* and *Pseudomonas aeruginosa* in environments with shear stresses, host defence, and low iron levels, such as the urinary tract [16]. In addition, it has also been shown that in cases of uncomplicated cystitis, biofilm formation on the uroepithelium is prevalent [18]. The uroepithelium also provides a physical barrier between the bladder contents and the underlying muscle and nerves, making it the primary site of infection by bacteria invading the lower urinary tract [19]. 

With respect to UPEC, their ability to invade bladder epithelial cells (BECs) has been identified as a key event that contributes to the persistence and recurrence of UPEC UTIs, given that it provides a reservoir of bacteria to cause recurrent infection episodes [20]. UPEC is known to do so by adhering to fusiform vesicles (vesicles that contain an additional membrane required for bladder expansion) that are internalised into superficial BECs, which is the initial stage of uroepithelium colonization [21]. 

Following invasion, UPEC form IBCs containing metabolically efficient daughter cells, which aid in bacterial evasion of host innate defences [22,23]. Internalised bacteria survive in these IBCs because they are unable to be destroyed by the immune response activated by bladder epithelial cell signalling in the presence of active infection [24,25]. Instead, the uroepithelium constantly undergoes turnover and differentiation, resulting in the shedding of these infected cells [23]. This causes expulsion of bacteria from the IBC reservoir, which results in bacteria reverting to an actively replicating form and, consequently, potentially causing a recurrent infection of the urine (bacteriuria) [26]. In addition, the progeny of IBC-internalised bacteria can often successfully invade and replicate in neighbouring cells within hours, thereby providing an additional survival advantage [27]. 

Recent research has investigated the effect of thiol antioxidants such as glutathione as an antibiofilm agent against ESKAPE pathogens [28]. Similarly, it has been established that N-acetylcysteine (NAC) plays a pivotal role in protecting cells from oxidative stress by neutralising free radicals and reacting with reactive oxygen species (ROS) [29]. The antibiofilm properties of NAC are also well established [18,30,31]. NAC acts as a potent antibiofilm agent alone and in combination with antibiotics such as fosfomycin and Augmentin against *Staphylococcus aureus* (methicillin-resistant and sensitive), *E. coli*, *E. faecalis*, *P. aeruginosa, Burkholderia spp.*, and other species, including surface-associated biofilms [18,32,33,34]. A recent insightful study showed that NAC penetrates the biofilm matrix and kills bacteria within, resulting in an empty gel-like matrix [35]. The researchers postulated that NAC destroys bacterial cells by dissociating and acidifying the bacterial cytoplasm, thereby causing protein and DNA degradation [35]. At lower concentrations of around 2 g/L, NAC alone can inhibit *E. coli* biofilm synthesis by 20–40% [18], and it can affect biofilm viability in *S. aureus* MRSA and MSSA by over 90% at concentrations of 4 g/L [32]. 

NAC has been widely investigated as an adjuvant for antibiotic treatment of various infections such as chronic bronchitis, vascular-related catheter infections, and urinary tract infections [18,36]. However, it is currently unknown if NAC provides specific protection against UTIs in the bladder and whether it is effective against established bladder infections.

This study aimed to investigate the efficacy of NAC as an antibiofilm agent against clinical uropathogenic *E. coli* and *E. faecalis* (refer to Table 1 for isolates used), the ability of NAC to inhibit the infiltration of bladder epithelial cells by uropathogens and prevent subsequent intracellular biofilm formation, the cytotoxicity of NAC on BECs at its intrinsic pH, and the effect of bacterial inflammatory stimulation following the treatment of cells. 

## 2. Results

### 2.1. NAC Inhibits Uropathogenic Growth and Biofilm Formation

Except for one *E. coli* strain, over 90% inhibition of planktonic growth was observed using NAC concentrations greater than 4.5 g/L and less than 8.5 g/L across all remaining tested *E. coli* and *E. faecalis* strains (Figure 1A,B; EC: *E. coli*, EF: *E. faecalis*), compared with untreated controls (normalised to 100% growth) (*p* < 0.0001 for all *E. coli* and *E. faecalis* isolates treated with 8.19 g/L compared with untreated controls). Significant biofilm inhibition was observed in all strains studied: >80% inhibition was noted in all *E. faecalis* strains and two *E. coli* strains, while EC AM-2 showed approximately 50% inhibition in biofilm formation. When NAC was combined with 1× MIC and ~1.5× MIC ciprofloxacin, a similar decrease in inhibition was observed (Figure 1C,D) (*p* < 0.05, 0.01). All bacterial strains also had a complete antibiotic sensitivity profile conducted (Table 1).

### 2.2. Bacterial Adhesion and Aggregation on BECs Are Inhibited in the Presence of NAC

A significant 7.5- to 10-fold decrease in bacterial counts/mm^2^ were recorded across all six strains when treated with ≥4.89 g/L NAC (*p* < 0.0001) (Figure 2H,I). For *E. faecalis* strains, the difference was most notable for the *E. faecalis*-2 strain, where there was an approximately 200-fold decrease from approximately 3700 cells/mm^2^ to 18 cells/mm^2^ when exposed to 8.16 g/L. Bacterial aggregation in the presence of bladder epithelial cells was also inhibited when exposed to 4.89 g/L and 8.16 g/L NAC overnight, as indicated by red arrows (Figure 2A–G). The morphology and confluency of BECs were also maintained and appeared identical to uninfected cells despite the inhibition of bacterial replication and aggregation, as is evident in the untreated controls of cells infected by *E. coli* and *E. faecalis* (Figure 2A,D, respectively).

### 2.3. Treatment with NAC or NAC + Ciprofloxacin Decreased Bacterial Burden in Biofilms

Mature *E. coli* biofilms treated with NAC alone resulted in an average decrease of 3_log10_ in CFU/mL (from 10^9^ to 10^6^ CFU/mL) and a significant 3–6_log10_ decrease in CFU/mL when combined with 0.025 g/L or 0.03 g/L ciprofloxacin (*p* < 0.0001 compared with untreated controls) (Figure 3A). A significant 4-log_10_ decrease was noted in colony counts of treated mature *E. faecalis* biofilms when treated with NAC + ciprofloxacin compared with untreated controls (Figure 3B). There was an insignificant 1–2_log10_ difference in the decrease in CFU/mL between NAC alone and NAC + ciprofloxacin. Additionally, a 3–4_log10_ decrease in CFU/mL with no significant difference in bacterial load was noted between 4.81 g/L NAC and 8.19 g/L NAC. A 1–2_log10_ decrease in CFU/mL was noted between treatment with 4.57 g/L and 4.89 g/L NAC, albeit not significant. No significant concentration-dependent effect was noted for either NAC or ciprofloxacin when combined. 

### 2.4. NAC + Ciprofloxacin Significantly Disrupted E. coli and E. faecalis Biofilm Biomass 

*E. coli* biofilm biomass showed significant disruption when treated with a combinaton of 4.81 g/L NAC and 0.03 g/L ciprofloxacin (Figure 4A–E). Untreated biofilms contained a majority of live biomass (~85%) (Figure 4A). Treatment with either 4.81 g/L or 8.16 g/L NAC alone rendered around 50% calculated dead biomass (Figure 4B,C). When combined, an average of 75% of the biomass died (*p* < 0.01) (Figure 4K). A similar trend was also observed with the treatment of *E. faecalis* biofilms. Untreated biofilms contained live biomass of approximately 85% (Figure 4F). This was significantly reduced to less than 50% live biomass when treated with 4.81 g/L NAC alone in comparison with the untreated controls and treatment with ciprofloxacin alone (Figure 4G). Treatment with 0.03 g/L ciprofloxacin alone only killed <10% of biofilm populations (Figure 4I). However, NAC + ciprofloxacin treatment (4.89 or 8.16 g/L NAC combined with 0.03 g/L ciprofloxacin) resulted in a significant killing of biomass (circa 65%) compared with treatment with ciprofloxacin alone (*p* < 0.01) (Figure 4L). 

### 2.5. NAC Does Not Display a Concentration-Dependent Cytotoxic Effect on BECs

BECs exposed to various concentrations of NAC maintained healthy cell viability (>85%) up to the 48 h time point (Figure 5A). Lethal Dose 50 (LD_50_) was only achieved closer to the 72 h time point with exposure to either 4.81 g/L or 8.16 g/L, while treatment with lower a concentration (1.63 g/L NAC) had no effect on cellular viability. Cytopathic effects were not visible until the 72 h time point with a concentration of 8.16 g/L (Figure 5B), corresponding to the point at which LD_50_ was achieved. Advanced cytopathic effects were visible at the 96 h time point, and most cells had a rounded shape and were nonviable. 

### 2.6. NAC Protects BECs from Bacterial Invasion In Vitro

The intracellular bacterial burden of BECs infected with *E. coli* showed a significant 4- to 5-fold average decrease when infected cells were treated with 8.16 g/L NAC (*p* < 0.01). Treatment with 4.61 g/L NAC resulted in a similar decrease in the intracellular bacterial load. Of the total bacterial burden of 10^8^ CFU/mL added to the cell monolayer, only around 50% (~10^5^ CFU/mL) was internalised in the untreated control. However, treatment with either 4.81 g/L or 8.16 g/L NAC resulted in no intracellular bacteria being recovered, apart from one exception wherein *E. coli* AM-2 was recovered when treated with 4.61 g/L NAC (Figure 6A). Intracellular bacteria were also visually observed when BECs were infected with a GFP tagged*-E. coli* AM-1 strain (Figure 6C–H). This showed internalised bacteria 2 h post-infection in the untreated control compared with no internalised bacteria observed with 4.81 g/L or 8.16 g/L NAC treatment. A similar pattern was observed in the internalisation of *E. faecalis* by BECs, for which 4.81 g/L NAC resulted in a 3 to 5-fold decrease in the bacterial load compared with untreated controls (from 10^5^ CFU/mL to 10^2^ CFU/mL, 10^3^ CFU/mL to <10^1^ CFU/mL, and 10^5^ CFU/mL to 0 for *E. faecalis*-1, 2, and 3, respectively). There was a significant decrease in the case of *E. faecalis*-2 (*p* < 0.05). A significant decrease (no bacteria recovered) was recorded when EF-3 was treated with 8.16 g/L NAC (*p* < 0.01) (Figure 6B).

### 2.7. NAC Damaged Bacterial Membranes and Decreased Polysaccharide Production in a Strain-Dependent Manner

Polysaccharide production in *E. coli* clinical isolates was affected in a strain-dependent manner, by 4.81 g/L and 8.16 g/L NAC, with *E. coli* AM-3 showing a significant decrease (almost 5-fold) in biofilm polysaccharide production (from 0.7 mg/mL to 0.1 mg/mL, *p* < 0.01) (Figure 7A). The effect on polysaccharide production in the other two *E. coli* strains was variable. For *E. faecalis*, isolates 2 and 3 underwent a significant decrease in polysaccharide production (Figure 7B). This production was more than halved (0.40 mg/mL to 0.199 mg/mL and 0.186 mg/mL) in *E. faecalis*-2 when treated with either 4.81 g/L or 8.16 g/L NAC, respectively (*p* < 0.01 compared with untreated controls). In *E. faecalis*-3, an almost 50% decrease in polysaccharide production was noted when treated with 8.16 g/L compared with the untreated controls (0.18 mg/mL) (*p* < 0.05). *E. faecalis*-1 showed a negligible difference when treated with NAC. Bacterial cells showed increased membrane damage (measured as an increase in propidium iodide (PI)-positive populations) when treated with NAC at both the 3 and 24 h exposure times. In *E. coli* AM-1, there was already a 2.5- to3-fold increase in PI-positive bacterial populations at the 3 h time point when treated with 4.81 g/L and 8.16 g/L NAC, respectively (Figure 7C). At 24 h, this increased to a significant 4- to 4.5-fold increase in PI-positive bacterial populations, directly corresponding to increased membrane damage (*p* < 0.05 compared with untreated controls). A very similar pattern was also observed in *E. faecalis*-3 populations stained with PI post-treatment, with a nearly fourfold increase in PI-positive populations 3 h post-treatment when treated with 4.81 mg/mL or 8.16 mg/mL NAC (Figure 7D). At the 24 h timepoint, an almost sixfold increase in PI-positive populations was noted when treated with 8.16 mg/mL NAC, and a nearly fivefold increase was noted when treated with 4.81 mg/mL NAC. Both were significantly higher compared with untreated controls (*p* < 0.05).

## 3. Discussion

NAC has been investigated for its biofilm disruption and prevention of biofilm formation properties for several decades. Its antibiofilm activity is hypothesised to stem from interactions between its thiol group (-SH) and bacterial cell wall proteins [30]. This current study demonstrated significant inhibition of biofilm formation by the uropathogens UPEC and *E. faecalis* when treated with NAC alone or in combination with ciprofloxacin. NAC and NAC + ciprofloxacin was also shown to significantly disrupt preformed biofilms. Treatment with NAC alone and in combination with ciprofloxacin resulted in a >3_log10_ decrease in CFU/mL, thus showing that NAC can efficiently disrupt preformed biofilms and enhance the killing of biofilm populations within the matrix. A significant drop in intracellular bacterial loads was also observed in the presence of NAC following infection with *E. coli* and *E. faecalis*. The bacterial loads recovered were consistent with the visualisation of cells infected with fuGFP-tagged *E. coli* (the same clinical isolate) in the presence of NAC, where no IBCs were observed 2 h post-infection. This is a landmark finding as it alludes to the idea that NAC prevents initial bladder cell invasion and potentially prevents subsequent intracellular bacterial replication and IBC reservoir formation. When internalised into the bladder urothelium, uropathogenic species often mature into biofilm states and form a reservoir that results in recurrent UTI episodes [37]. This is a cause for concern as bacterial loads can remain at a high level for almost 7–10 days, despite a strong neutrophil influx triggered by recognition of LPS by the bladder epithelium via the Toll-like receptor 4 (TLR-4)-CD-14 pathway [37]. A single uroepithelial cell can often contain 10^5^ CFU of bacteria and represent a single IBC [38]. While the invasion–inhibition effect of NAC is concentration-dependent, our findings demonstrate treatment with NAC is likely to prevent the formation of bacterial reservoirs that could potentially cause recurrent episodes of UTIs.

The effects of NAC on cellular entry by uropathogenic bacteria were investigated for the first time in this study. Our results showed that NAC attenuates bacterial entry and subsequent establishment in bladder epithelial cells. Gamaley et al. (2006) attempted to define the mechanism by which NAC prevents invasion using mouse embryonic fibroblast cells [39]. They concluded that despite increased intracellular glutathione levels in cells treated with 20 mM (2.78 g/L) NAC, increased GSH was not the mechanism by which cell invasion was prevented [39]. Instead, they suggested that NAC could potentially induce synthesis or modification of proteins that conceal recognition of cell membranes by bacteria by interacting with surface membrane proteins through their sulfhydryl groups, which is usually the first step in cell–bacterial interactions [39]. Whether this is the case with bladder cells and the invasion of uropathogens is yet to be determined. 

Ciprofloxacin is commonly used against UTIs given its activity against various Gram-negative pathogens, high urinary excretory rate, and safety profile [40]. However, recent studies have highlighted an increasing resistance to ciprofloxacin in UPEC strains worldwide [40]. Combining ciprofloxacin with a disruptive bacteriostatic agent such as NAC increases individual cell exposure to ciprofloxacin, and our study showed that it resulted in a significantly greater loss of bacterial load compared with ciprofloxacin alone. Previous studies have investigated the significant potential of a NAC/ciprofloxacin combination or NAC alone to inhibit catheter colonisation and subsequent biofilm formation by uropathogens, demonstrating that bacterial loads were reduced by more than 90% [41,42]. Our study further expanded on these findings by showing similar reductions in non-catheter-associated uropathogen growth and demonstrating the inhibition of Gram-positive and -negative bacterial entry into BECs when treated with NAC alone. 

The ability of BECs to withstand the very acidic pH of NAC (approximately 2.5) for up to 72 h (Figure 5) is also highly advantageous to the in vivo use of this antioxidant, as biofilm formation in the bladder or IBC formation can be directly targeted without further drug-induced cytotoxicity. This is a key finding, as it indicates that a combination of NAC and an antibiotic could potentially be used as direct bladder irrigation to treat recurrent UTIs. Research has shown a reduction in symptom frequency and antibiotic usage when bladder irrigation is employed in the treatment regimen for chronic UTIs [43]. BECs undergo rapid shedding when infected [44], exposing the immature bladder epithelium to pathogenic infiltration and consequent formation of quiescent intracellular reservoirs [44]. As we demonstrated, NAC prevents bacterial cell invasion in vitro, this could potentially also reduce further dissemination of an infection during a UTI episode and prevent increased bacterial shedding during recurrent episodes. Our in vitro assay results on BEC cell lines validated previous in vivo and in vitro findings of epithelial cell invasion by uropathogens [20]. However, further investigations using in vivo UTI mouse models are required to identify if this invasion protection is exercised by NAC.

This study also showed that NAC degraded secreted and matrix-bound polysaccharides in a strain-dependent manner in the uropathogens tested. This finding has also been observed in other biofilm-forming pathogens, such as *S. aureus* MSSA and MRSA, *P. aeruginosa*, and *Helicobacter pylori* [32,45,46]. Degrading polysaccharides required for matrix formation could affect the adhesive properties of the matrix to a surface and subsequent robust biofilm formation [36], the characteristics of which are especially vital for bacterial survival in the urinary tract given the shear stress bacteria are exposed to by urine flow forces [47]. 

When given alone, many antibiotics prescribed for UTIs, including ciprofloxacin, fail to eliminate the intracellular uropathogenic burden in mouse models [48]. This has led to the suggestion that recurrent UTIs in patients potentially stem from intracellular quiescent reservoirs not eradicated by antibiotic therapy [48]. Biofilm formation is also induced in many bacterial species in the presence of antibiotics [49], suggesting a role in the upregulation of resistance factors [50]. Our study showed that not only is NAC effective in reducing these intracellular reservoirs such as within IBCs, but there was no antagonistic activity between NAC and ciprofloxacin. Combining NAC with ciprofloxacin enhanced bacterial killing during treatment. Thus, the combination of an antibiotic with a stable antioxidant such as NAC could be the most effective way to treat urinary tract infections and prevent their recurrence. 

## 4. Materials and Methods

### 4.1. Reagents and Media Used in This Study

NAC, ciprofloxacin, trypsin-EDTA, gentamicin, Triton X-100, sodium chloride, resazurin, 1× phosphate-buffered saline (PBS), glycerol, 89% (*v*/*v*) phenol, and sulphuric acid were purchased from Sigma-Aldrich, Australia. Tryptic soy broth (TSB) and Tryptic soy agar (TSA) were obtained from Oxoid (Thermo Fisher, Scoresby, Australia). RPMI 1640 media containing L-glutamine, sodium bicarbonate, and penicillin/streptomycin solution were also obtained from Sigma-Aldrich. Foetal Bovine Serum (FBS) was purchased from Thermo Fisher. Alexa Fluor^®^ 647 conjugate of Wheat Germ Agglutinin (WGA) and 4′,6-diamidino-2-phenylindole (DAPI) nucleic acid stain were obtained from Thermo Fisher.

### 4.2. Cell Culture

The human bladder epithelial cell line 5637 (ATCC 5637) stocks were stored in liquid nitrogen in Foetal Bovine Serum FBS containing 20% (*v*/*v*) Dimethylsulfoxide (DMSO). For all downstream uses, cell stocks were resuscitated and cultured in RPMI 1640 (Sigma-Aldrich, Australia) media supplemented with 10% (*v*/*v*) FBS and 1% (*w*/*v*) penicillin/streptomycin at 37 °C in a humidified atmosphere containing 5% (*v*/*v*) CO_2._ For cytotoxicity and invasion assays, RPMI media supplemented with 10% (*v*/*v*) FBS was used.

### 4.3. Determining the Effect of NAC on Planktonic Growth of UPEC and E. faecalis

All bacterial strains were grown overnight in TSB at 37 °C and 150 rpm orbital shaking. Cultures were then centrifuged at 5000× *g* for 5 min at 10 °C and the pellet was resuspended in fresh TSB. Bacterial cultures were inoculated (*t* = 0 h) at OD_600_ = 0.1 ± 0.02 in 96-well, flat-bottomed plates (Corning Corp., Corning, NY, USA) in the presence of 3.26, 4.89, 5.71, 6.53, and 8.16 g/L NAC at *t* = 0. Plates were incubated for 48 h at 37 °C and 100 rpm, and OD_600_ was recorded at *t* = 48 h using a plate reader (Tecan Infinite M1000 pro, Melbourne, Australia).

### 4.4. Determining Minimum Biofilm Inhibitory Concentrations for Uropathogenic Strains

Overnight bacterial cultures at a density of OD_600 nm_ = 0.1 ± 0.01 were added to 96-well tissue culture plates in PBS and allowed to adhere for 1 h at 37 °C, 100 rpm. Wells were gently washed once with 1× PBS to remove any unadhered bacteria. TSB containing varying concentrations of NAC was then added, and the plates were incubated for an additional 48 h. Biofilm formation was measured at *t* = 48 h by measuring bacterial density at OD_600 nm_.

### 4.5. Investigating the Effect of NAC on Initial Adhesion of Uropathogens to Polymer Substratum

The effect of NAC on the initial adhesion of UPEC and *E. faecalis* was investigated as mentioned previously [32]. Briefly, six-well plates (Corning Corp., Corning, NY, USA) were seeded with bacteria at a density of OD_600 nm_ = 0.1 ± 0.02 and incubated for 2 h at 37 °C and 100 rpm. Plates were then washed twice with 1× PBS and imaged using phase contrast microscopy at 200× magnification (Zeiss AxioScope.A1 FL, LED, Jena, Germany). 

### 4.6. Quantification of Bacterial Load in Biofilms Exposed to Treatment

Preformed biofilms were grown by inoculating 96-well, flat-bottomed plates with overnight bacterial cultures at OD_600_ = 0.1 ± 0.02. Plates were then incubated for 48 h in TSB at 37 °C and 100 rpm to allow for biofilm growth. Biofilms were then washed once with 1× PBS and treated as described (Table 2) for 24 h. Following incubation, the treatment was removed, and the biofilms were washed once with 1× PBS and then scraped from the surface and homogenized in 200 µL 1× PBS. Serial dilutions were then plated onto TSA plates and incubated at 37 °C overnight. Colonies were enumerated and expressed as CFU/mL.

### 4.7. Measuring NAC Cytotoxicity on Bladder Epithelial Cells (BECs)

To measure the cytotoxicity of NAC on BECs, six-well plates were seeded with cells at a density of 1 × 10^6^ cells/mL and allowed to grow to confluence for 48 h. Cells were then washed with RPMI media once and treated with concentrations of NAC while incubated at 37 °C in a humidified atmosphere of 95% (*v*/*v*) air and 5% (*v*/*v*) CO_2_. The NAC was removed, and cells were washed once with RPMI media to remove any residual NAC. The plates were imaged using phase contrast microscopy (Zeiss AxioScope.A1 FL LED, Jena, Germany) at 200× magnification. To measure cellular metabolic viability, cells were stained with 0.05% *w*/*v* resazurin in RPMI media for 24 h, and fluorescence intensity was measured using a plate reader (Tecan Infinite M1000 pro; Ex/Em at 544/590 nm).

### 4.8. Determining Inhibition of Bacterial Growth in Presence of NAC on Pre-Confluent BECs 

BECs were seeded in 12-well tissue culture plates at a density of 5 × 10^5^ cells/mL and allowed to grow to 95% confluence. Cells were inoculated with bacteria (multiplicity of infection of 10) in the presence of NAC, and plates were incubated at 37 °C and 5% (*v*/*v*) CO_2_ for 24 h. Plates were then washed twice to remove non-adhered bacteria and subsequently imaged using phase contrast microscopy (Zeiss AxioScope.A1 FL, LED, Jena, Germany) to identify bacterial aggregation.

### 4.9. GFP-Tagging of Clinical Uropathogenic E.coli for Confocal Microscopy Tracking of Uropathogenic Invasion

Clinical *E. coli* was tagged with green fluorescent protein (GFP) following a previously published protocol with free use GFP b (fuGFPb) [51]. *E. coli* cells were made electrocompetent by centrifuging overnight cultures of *E. coli* AM-1 (5000× *g*, 4 °C for 15 min). The supernatant was discarded and the pellet was resuspended in 50 mL of cold sterile 10% *v*/*v* glycerol and centrifuged (4500 rpm, 4 °C for 15 min). The pellet was then resuspended in 5 mL of 10% (*v*/*v*) glycerol again to make bacteria electrocompetent and stored at −80 °C prior to transformation. Plasmid DNA containing fuGFPb and kanamycin resistance was added to 50 μL of bacteria and the mixture was transferred to an electroporation cuvette. The mixture was subjected to 2500 V, 25 µF, and 200 Ω on the electroporator (Gene Pulser Xcell Microbial System, BIO-RAD, Hercules, CA, USA), and then incubated for 1 h at 37 °C under dynamic conditions. Following this, the transformed bacteria were plated onto Luria–Bertani (LB) agar containing kanamycin to select for successfully transformed bacterial colonies. 

### 4.10. Determining Inhibition of Bacterial Cell Invasion in Presence of NAC

BECs were seeded in 12-well tissue culture plates at a density of 5 × 10^5^ cells/mL and allowed to grow to 95% confluence. Cells were washed once with RPMI media and exposed to NAC 2 h prior to infection. Overnight grown bacterial cultures were centrifuged and bacterial pellet resuspended in RPMI supplemented with 10% (*v*/*v*) FBS to an OD_600_ = 0.1 ± 0.02. Bladder cells were then infected with bacteria at a multiplicity of infection (MOI) of 10 for 2 h in the presence of NAC. The bladder cells were washed to remove any non-adherent bacteria and treated with 1 mg/mL gentamicin for 1 h to remove any extracellular bacteria. Cells were then lysed with 0.5% (*v*/*v*) Triton-X 100 and serial dilutions plated onto TSA plates to enumerate intracellular bacterial loads, expressed as CFU/mL.

### 4.11. Visualising Inhibition of Uropathogenic E.coli Intracellular Bacterial Colonies in BECs

To provide visual support to the invasion assay above, our clinical model strain (*E. coli* AM-1) was tagged with fuGFP (as mentioned above) to track its intracellular bacterial colony formation using fluorescence microscopy. Cells were seeded in a 4-well chamber slide at a density of 10,000 cells/mL and incubated for 48 h to grow to confluence. Cells were washed once with 1× PBS and infected with fuGFP tagged *E. coli* following the invasion assay protocol described above. The cells were stained with Alexa fluor-647 conjugated WGA (Invitrogen, Thermo Fisher, Melbourne, Australia) for 10 min to stain the cytoskeleton (WGA binds to glycoproteins of the cell membrane), and then counterstained with DAPI (Invitrogen, Thermo Fisher, Melbourne, Australia) for 1 min. Slides were fixed with 10% paraformaldehyde and imaged using a Confocal Laser Scanning Microscopy (CLSM, Olympus FV1200, Melbourne, Australia).

### 4.12. Establishing Inhibition of Exopolysaccharide Production by NAC in Biofilm Matrix

To investigate if NAC inhibits exopolysaccharide production in uropathogenic biofilms, polysaccharides from the biofilm matrix were extracted and quantified as previously described by Chiba et al. (2016) [52]. In brief, biofilm formation in the presence of NAC was initiated by inoculating 10 mL of TSB containing various concentrations of NAC with OD_600_ = 0.1 ± 0.02 overnight bacterial cultures in 15 mL Falcon tubes. The tubes were incubated at 37 °C under static conditions for 24 h and subsequently centrifuged at a low speed (20 min at 2851× *g* and 25 °C) to pellet only the heavier biofilm material. The supernatant was discarded, and the pellet was resuspended in 10 mL of 1× PBS, followed by centrifugation at 2851× *g* for 20 min to remove NAC. A second wash in 10 mL of 1× PBS was then performed, followed by centrifugation again to eliminate the effects of residual NAC. The pellet was resuspended in 100 μL of NaCl, followed by centrifugation (4713× *g* for 10 min). The resulting supernatant, comprising the extracted extracellular polymeric matrix (EPM), was transferred into 1 mL Eppendorf tubes. Phenol at a concentration of 5% (*v*/*v*) and pure sulphuric acid was then added to 10 μL of EPM, following the Dubois method [53]. The colour change obtained after 1 h incubation was measured at OD_550_ using a plate reader (Tecan Infinite M1000 pro, Australia) to quantify the polysaccharides present.

### 4.13. Disruption of Biofilm Architecture Following Treatment with NAC

Biofilms were grown by inoculating 4-chambered slides with 1 mL of overnight bacterial culture at OD_600_ = 0.1 ± 0.02 in TSB, and slides were incubated for 48 h at 37 °C under static conditions. Slides were gently washed once with 1× PBS to remove unattached bacteria and treated with varying concentrations of NAC and ciprofloxacin for 24 h. Biofilms were then washed once with 1× PBS and stained with a live/dead stain (Bacterial Viability Kit, Molecular Probes, Thermo Fisher Scientific, USA) followed by incubation in the dark for 45 min prior to imaging. Biofilms were visualized by CSLM (Olympus FV1200, Australia) with Ex/Em 473/559 nm and Ex/Em 500/637 nm for Syto 9 (green for live cells) and PI (red for damaged and dead cells) staining, respectively. All biofilms were imaged at 400× magnification with 3D proportions. Images were analysed and generated using Fiji ImageJ software [54]. 

### 4.14. Quantification of Bacterial Membrane Damage by NAC 

To quantify membrane damage of planktonic bacterial cells due to NAC, bacterial cultures at a density of OD_600 nm_ = 0.1 ± 0.02 were incubated in the presence of 1.69, 4.81, and 8.16 g/L of NAC at 37 °C, 100 rpm for 3 and 24 h. At the given time points, the bacterial cultures were centrifuged at 6500× *g* for 10 min to remove residual NAC. Cells were washed once with 1× PBS and resuspended in 1 mL 1× PBS. Treated bacterial cell cultures (100 μL) were dispensed into a 96-well black microplate (Corning Corp., USA), stained with 15 µM PI, and incubated for 1 h in the dark. Fluorescence readings were obtained using a microplate reader (Tecan Infinite M1000 pro) at Ex/Em 535/617 nm. 

### 4.15. Statistical Analysis of Data

One-way ANOVA followed by Tukey’s multiple comparisons test or the Kruskal–Wallis test was performed using GraphPad Prism 8.0.0 (GraphPad, San Diego, CA, USA) according to the controls of each data set to obtain statistical significance.

## Figures and Tables

**Figure 1 antibiotics-10-00900-f001:**
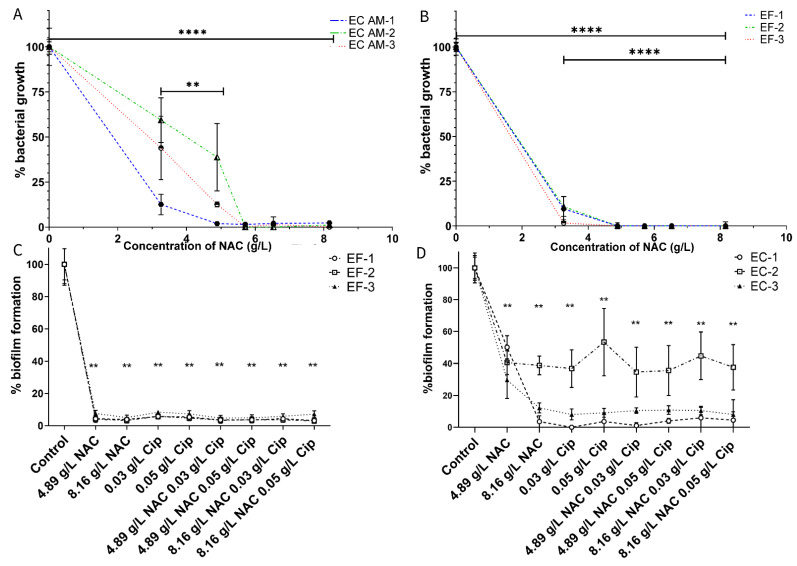
The inhibitory effect of NAC on planktonic growth and biofilm formation of uropathogenic *E. coli* and *E. faecalis.* A concentration-dependent inhibitory effect of NAC was observed at *t* = 48 h in three EF (*E. faecalis*) and three EC (*E. coli*) clinical strains (**A** and,**B**). **** *p* < 0.0001 compared with untreated controls; ** *p* < 0.001 compared with lower concentrations using Tukey’s multiple comparisons test (**C**,**D**) display the minimum biofilm inhibitory concentrations of NAC and NAC + ciprofloxacin EF and EC, respectively. ** *p* < 0.001 compared with lower concentrations using Tukey’s multiple comparisons test. Cip: ciprofloxacin in graphs. Data represent the means ± SD of *n* = 5 biological replicates.

**Figure 2 antibiotics-10-00900-f002:**
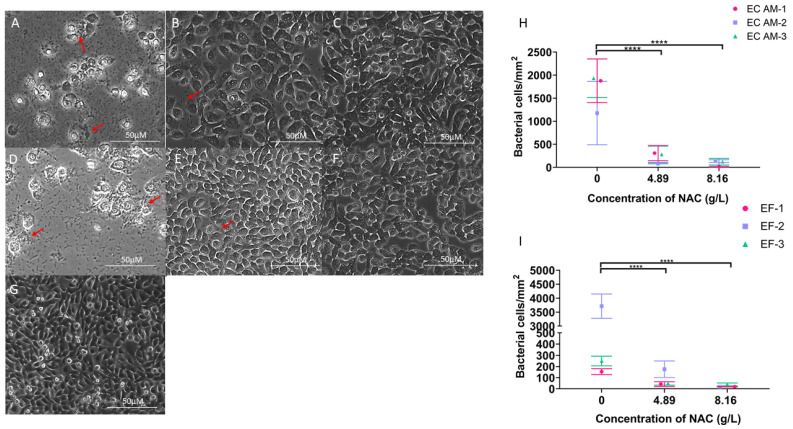
NAC prevented bacterial adhesion to the substratum and bacterial aggregation in presence of bladder epithelial cells (BECs). (**A**–**G**) show absence of bacterial aggregation in infected bladder epithelial cells in the presence of NAC. (**A**): Bladder epithelial cells in presence of EC (*E. coli*)-AM 1 alone. (**B**): Infected BECs in presence of 4.89 g/L NAC. (**C**): Infected BECs in presence of 8.16 g/L NAC (**D**): Bladder epithelial cells in the presence of EF (*E. faecalis*)-3 alone. (**E**): EF-3-infected BECs in the presence of 4.89 g/L NAC. (**F**): EF-3-infected bladder epithelial cells in the presence of 8.16 g/L NAC. (**G**): Uninfected confluent bladder epithelial cells. (**H**): Enumeration of *E. coli* bacterial cells attached to the substratum during initial adhesion in the presence of varying concentrations of NAC. Red arrows indicate bacterial aggregation. (**I**): Enumeration of *E. faecalis* bacterial cells attached to the substratum during initial adhesion in the presence of varying concentrations of NAC. Data represent the means ± SD of *n* = 5 biological replicates. **** *p* < 0.0001 compared with untreated controls using Tukey’s multiple comparisons test.

**Figure 3 antibiotics-10-00900-f003:**
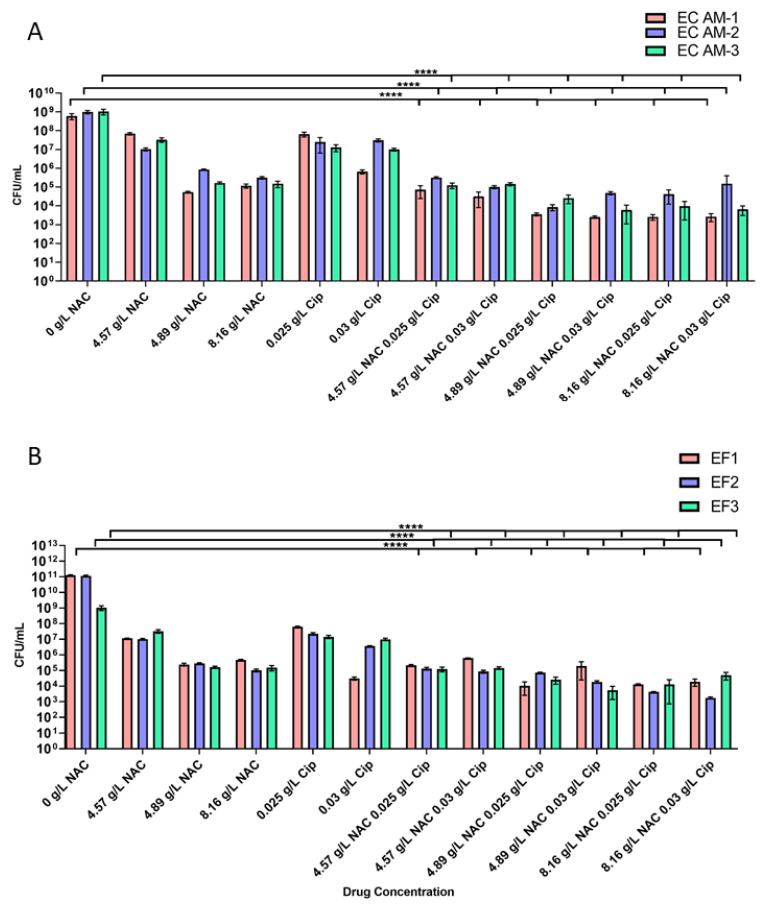
Quantification of the decrease in bacterial loads in NAC and NAC + ciprofloxacin-treated biofilms. The figure shows the concentration-dependent decrease in bacterial loads in uropathogenic EC (*E. coli*) strains (**A**) and EF (*E. faecalis*) strains (**B**), following treatment with NAC alone and NAC + ciprofloxacin. **** *p* < 0.0001 compared with untreated controls using Tukey’s multiple comparisons test. Data represent means ± SD of *n* = 4 biological replicates.

**Figure 4 antibiotics-10-00900-f004:**
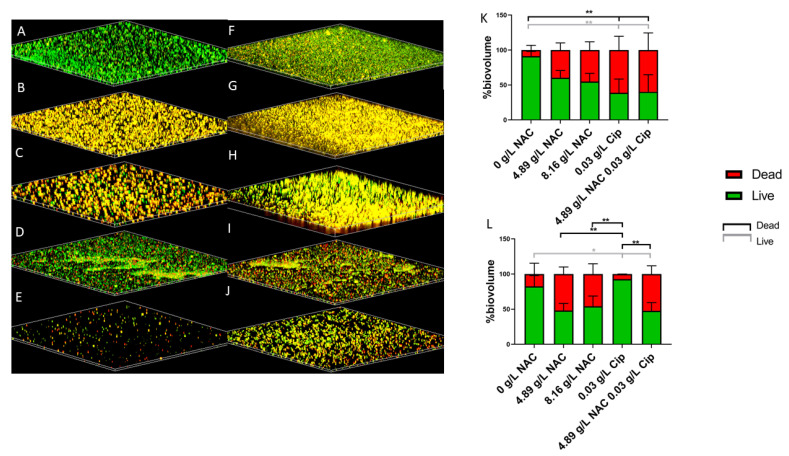
Visualisation of disruption in the *E. coli* and *E. faecalis* biofilm architecture after treatment with NAC and NAC + ciprofloxacin. The disruption caused to *E. coli* biofilms (**A**–**E**) and *E. faecalis* biofilms (**F**–**J**) resulting from treatment with various concentrations of NAC and NAC + ciprofloxacin is evident as dark regions in the Confocal Scanning Laser Microscope scan. (**A**–**E**): *E. coli* AM-1 biofilms. (**A**): Untreated. (**B**): Treated with 4.18 g/L NAC, (**C**): 8.19 g/L NAC, (**D**): 0.03 g/L ciprofloxacin, and (**E**): 4.81 g/L NAC + 0.03 g/L ciprofloxacin. (**F**–**J**): *E. faecalis*-3 biofilms. (**F**): Untreated. (**G**): Treated with 4.18 g/L NAC, (**H**): 8.19 g/L NAC, (**I**): 0.03 g/L ciprofloxacin, and (**J**): 4.81 g/L NAC + 0.03 g/L ciprofloxacin. (**K**): Quantification of the percentage of live/dead biovolume in *E. coli* AM-1 biofilm. (**L**): Quantification of the percentage of live/dead biovolume in the EF-3 biofilm. Images are representative of all *E. coli* and *E. faecalis* strains tested in this study. Data represent the means ± SD for *n* = 3 biological replicates. Tukey’s multiple comparisons test was used for statistical analysis (** *p* < 0.01, * *p* < 0.05). Black bars compare the significance between dead biovolume and grey bars compare the significance between live biovolumes.

**Figure 5 antibiotics-10-00900-f005:**
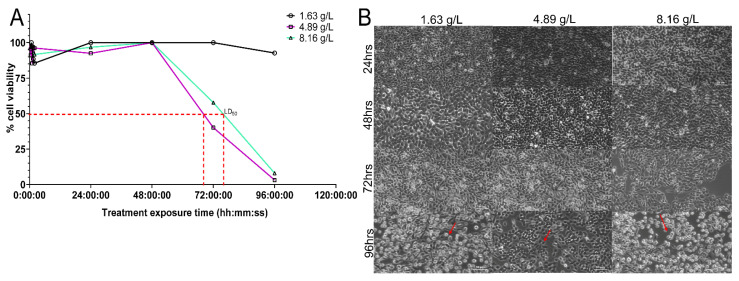
NAC does not exhibit cytotoxicity towards bladder epithelial cells at intrinsic pH. Cytotoxicity of NAC on 5637 BECs was tested at various concentrations for an exposure period of 96 h. BECs were exposed to NAC at various time points (at 2 h intervals for the first 6 h and then at 24 h intervals). At each timepoint, wells were washed and imaged. Following imaging, resazurin was added to measure cellular viability. (**A**) displays the effect of varying concentrations of NAC (1.63–8.16 g/L) on cellular metabolic activity over a period of 96 h. The vertical dotted lines indicate a lethal dose of 50% at which only 50% of cellular metabolism was sustained. (**B**) shows a panel of images taken using phase contrast microscopy at the specified timepoints correlating to viability measurements on their respective graphs in Figure 5A. The red arrows in 5B indicate any cytopathic effects observed. Data represent the means ± SD of *n* = 3 biological replicates.

**Figure 6 antibiotics-10-00900-f006:**
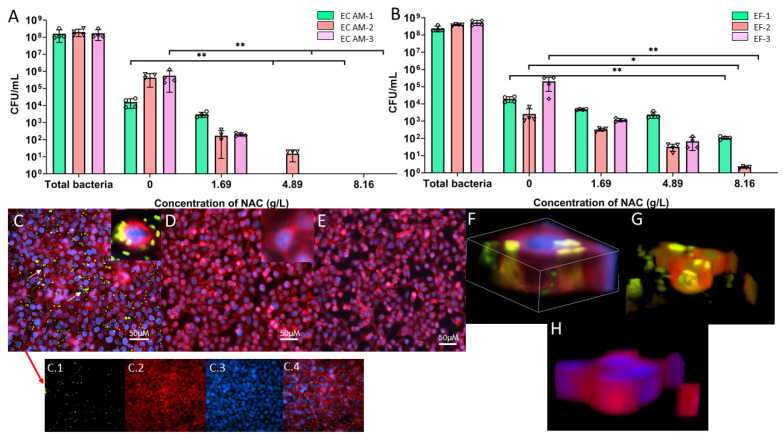
NAC protects cells from bacterial invasion and intracellular bacterial colony formation. Cells infected with either EC (*E. coli*) AM-1, AM-2, and AM-3, or EF (*E. faecalis*)-1, 2, and 3 in the presence of different concentrations of NAC exhibited a decreased intracellular bacterial load. (**A**) shows quantification of intracellular bacterial load recovered from cells infected with *E. coli* clinical strains and (**B**) shows the same with *E. faecalis* clinical strains. (**C**–**E**) display the visualised intracellular bacterial colony formation in 5637 BECs infected with free use GFP b (fuGFPb) tagged *E. coli* AM-1 and the absence of IBCs in cells treated with NAC. (**C**): Untreated controls. C.1, GFP excitation under FITC filter; 6C.2, WGA staining under Cy5 filter; 6C.3, DAPI staining under DAPI filter; C.4, all lasers overlayed. (**D**): Treated with 4.81 g/L NAC. (**E**): Treated with 8.16 g/L NAC. (**F**): Three-dimensional visualisation of invaded epithelial cells under all three filters. (**G**): Three-dimensional visualisation of invaded cells under FITC and Cy5 filters only. (**H**): Three-dimensional enlarged image of an infected epithelial cell treated with NAC under all three filters. Data represent the means ± SD of *n* = 4 biological replicates. The Kruskal–Wallis multiple comparisons test was used for statistical analysis. ** *p* < 0.01, * *p* < 0.05.

**Figure 7 antibiotics-10-00900-f007:**
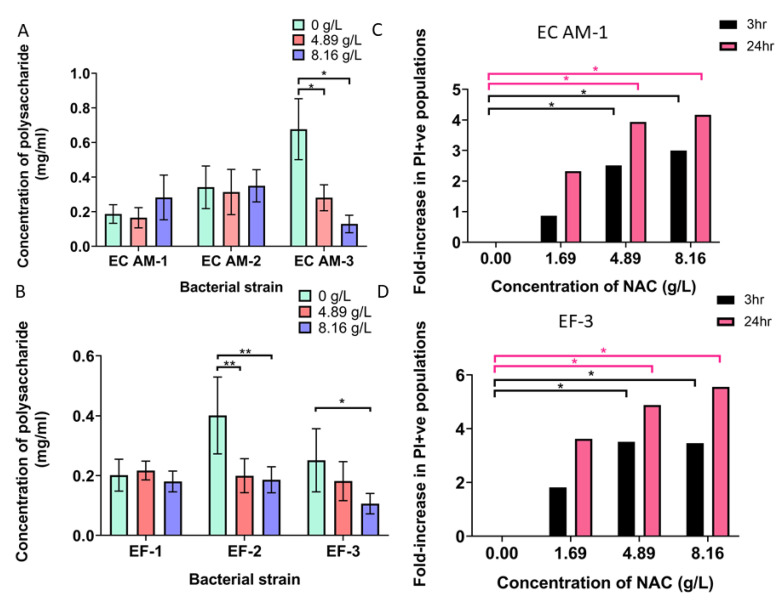
Treatment with NAC increases the proportion of damaged cell populations and affects matrix polysaccharide production. Figure 7 displays the effect of NAC on biofilm matrices and their contained bacterial populations. (**A**,**B**) show the strain-dependent effect of NAC on polysaccharide production in biofilm matrices of EC (*E.coli*) and EF (*E. faecalis*) respectively. (**C**,**D**) display the fold increase in dead *E. coli* and dead *E. faecalis* bacterial cell populations, respectively, at 3 and 24 h time points when exposed to NAC. Tukey’s multiple comparisons test was used for statistical analysis. * *p* < 0.05, ** *p* < 0.01. Data represent the means ± SD of *n* = 4 biological replicates.

**Table 1 antibiotics-10-00900-t001:** Bacterial strains used in this study, their origins, and their antibiotic sensitivity profiles. S, sensitive; R, resistant; NT, not tested; CIP, ciprofloxacin; AMK, amikacin; AMC, amoxicillin/clavulanate; NF, nitrofurantoin; TC, tetracycline.

Bacterial Species	Source	Antibiotics				
		CIP	AMK	AMC	NF	TC
***E. faecalis*** **-1**	St George Hospital, Kogarah, NSW, Australia; urine isolate	S	S	S	S	S
***E. faecalis*** **-2**	St George Hospital, Kogarah, NSW, Australia; indwelling catheter isolate	S	S	S	S	S
***E. faecalis*** **-3**	St George Hospital, Kogarah, NSW Australia; suprapubic catheter isolate	S	S	S	S	NT
***E. coli*** **AM-1**	Microbiology department Royal Prince Alfred Hospital (RPAH), Sydney, NSW, Australia; urine isolate	S	S	S	S	R
***E. coli*** **AM-2**	Microbiology department RPAH, Sydney, NSW, Australia; urine isolate	S	S	S	S	R
***E. coli*** **AM-3**	Microbiology department RPAH, Sydney, NSW, Australia; urine isolate	S	S	S	S	R

Isolate susceptibility was determined at the Microbiology Department, RPAH, Sydney, NSW, Australia and St George Hospital, Kogarah, NSW, Australia using the disc diffusion method, and breakpoints were determined following CLSI guidelines (CLSI-M100).

**Table 2 antibiotics-10-00900-t002:** Concentrations of NAC and ciprofloxacin tested alone and in combination on UPEC and *E. faecalis* strains and/or in cytotoxicity assessment.

Combinations of Treatment Tested	NAC (g/L)	Ciprofloxacin (g/L)
	1.63	-
	4.57	-
	4.89	-
	8.16	-
	-	0.025
	-	0.03
	4.57	0.025
	4.57	0.03
	8.16	0.025
	8.16	0.03

## Data Availability

Not applicable.

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
