# Peer review of "N-Acetylcysteine Protects Bladder Epithelial Cells from Bacterial Invasion and Displays Antibiofilm Activity against Urinary Tract Bacterial Pathogens"

_antibiotics, 2021, doi:10.3390/antibiotics10080900_

Round 1
Reviewer 1 Report
Overall I think the paper is well written and addresses an interesting emerging gap. There are a few small corrections I believe should be made:
1) Define IBCs line 65, as the first definition is later in the introduction
2) I'm not sure the current location is the best spot for Tables 1 and 2. Additionally both should be introduced in the text
3) Line 265, define PI (only defined in abstract prior)
Author Response
We would like to thank Editor and the reviewers for their significant comments on our manuscript (ID: antibiotics-1297681) entitled: “N-Acetylcysteine protects bladder epithelial cells from bacterial invasion and displays antibiofilm activity against urinary tract bacterial pathogens” which have helped us to strengthen the manuscript. Below we present point-by-point answers to all the reviewers’ comments. We have also uploaded the revised version of the manuscript and figures as well as our responses to reviewers in review forum. All changes in the manuscript are highlighted in yellow. I hope the manuscript will now be acceptable to you and the reviewers for publication in “Antibiotics”.
Yours Sincerely
Dr. Theerthankar Das, on behalf of all authors.
Response to the reviewer’s comments:
Reviewer 1
1) Define IBCs line 65, as the first definition is later in the introduction
Answers to reviewers: Thank you for pointing this out, we have added below sentence.
“however whether it forms intracellular bacterial communities (IBCs) (which are biofilm like communities)”. Changes highlighted in yellow. (current line 66-67).
------------------------------------
2) I'm not sure the current location is the best spot for Tables 1 and 2. Additionally both should be introduced in the text
Answers to reviewers: Thank you for your opinion on this. We have now introduced Table 1 in the introduction and moved it earlier in the text (refer line 117-118). Table 2 just displays concentrations tested hence we have left that where the material and methods are (Refer line 425-427).
-----------------------------------------------
3) Line 265, define PI (only defined in abstract prior)
Answers to reviewers: Thank you for pointing this out, we have now rectified and defined PI.
“increased membrane damage (measured as an increase in Propidium Iodide (PI) positive populations)”. Changes highlighted in yellow. (current line 277-278).

Reviewer 2 Report
This manuscript investigated whether NAC, alone or in combination with ciprofloxacin, can prevent uropathogenic E. coli and E. faecalis from invading bladder cells. The results showed that NAC is a non-toxic anti-biofilm agent in vitro and can prevent cell invasion and IBC formation by uropathogens, which may be a novel and effective treatment for UTI. I think it is worthy of publication because it is well written and the results are clear. The scientific names, such as E. coli and E. faecalis, are not italicized in many places, so that needs to be corrected.Author Response
We would like to thank Editor and the reviewers for their significant comments on our manuscript (ID: antibiotics-1297681) entitled: “N-Acetylcysteine protects bladder epithelial cells from bacterial invasion and displays antibiofilm activity against urinary tract bacterial pathogens” which have helped us to strengthen the manuscript. Below we present point-by-point answers to all the reviewers’ comments. We have also uploaded the revised version of the manuscript and figures as well as our responses to reviewers in review forum. All changes in the manuscript are highlighted in yellow. I hope the manuscript will now be acceptable to you and the reviewers for publication in “Antibiotics”.
Yours Sincerely
Dr. Theerthankar Das, on behalf of all authors.
Reviewer 2
1) This manuscript investigated whether NAC, alone or in combination with ciprofloxacin, can prevent uropathogenic E. coli and E. faecalis from invading bladder cells. The results showed that NAC is a non-toxic anti-biofilm agent in vitro and can prevent cell invasion and IBC formation by uropathogens, which may be a novel and effective treatment for UTI. I think it is worthy of publication because it is well written, and the results are clear. The scientific names, such as E. coli and E. faecalis, are not italicized in many places, so that needs to be corrected.
Answers to reviewers: Thank you for your comments. We have now updated the whole manuscript with italicized scientific names.

Reviewer 3 Report
The manuscript " N-Acetylcysteine protects bladder epithelial cells from bacterial invasion and displays antibiofilm activity against urinary tract bacterial pathogens " by Arthika Manoharan et al. is interesting. The work is well prepared. However, I have several comments to the work.
Page 2, lines 49-54.
“Uropathogenic Escherichia coli (UPEC) is the most prevalent UTI-causing bacterial species, accounting for over 90% of diagnosed community acquired and nosocomial UTIs6. These mainly arise from the colonisation of the periurethral mucosa by faecal bacteria that ascend through the urinary tract7,8. UPEC accounts for 80% and 33% of all uncomplicated and complicated infections, respectively9, 10. Other uropathogens such as Enterococcus spp. account for around 15% of all complicated UTIs in the hospital setting11.
Please analyze the quoted data. E. coli accounts for approximately 90% of community acquired UTIs, but the rates are much lower for hospital acquired UTIs.
Figure 1
Why did the Authors use a line chart to illustrate the obtained results?
In the Materials and Methods section, the Authors describe that bacterial cultures were incubated in the presence of 4.57, 4.81, 8.16 g / L of NAC (L 394), while the graph shows (Fig. 1A, 1B) that these values were different. Please explain.
Paragraph 2.1.
Imprecise interpretation of the results, e.g. grater than 4.5 g / L (i.e. 100g / L too?),> 80% inhibition,> 50% inhibition (80% is also more than 50%)
Page 3, line 128.
The Authors write that ciprofloxacin was tested at the concentrations of 1x MIC and 1.5xMIC. The graph (Figure 1) shows the values of 30 g/L and 50 g/L (50 g is not 1.5x30). In Table 2, the Authors indicated that the tested strains were sensitive to ciprofloxacin. According to EUCAST, the MIC breakpoint of ciprofloxacin for Enterobacterales is 0.25 mg/L, and for Enterococcus spp. 4 mg/L. How will the Authors explain these discrepancies?
Figure 2.
What do the red arrows indicate?
Paragraph 2.2.
Imprecise interpretation of results: “Six strains when treated with ≥4.89 g/L NAC”,
What does it mean: "there was a > 20 fold decrease from approximately 3700 cells / mm2 to 18 cells / mm2 when exposed to 8.16 g / L" 200 fold, 1000 fold?
Minor corrections
1) The names of the microorganisms should be italicized.
Author Response
We would like to thank Editor and the reviewers for their significant comments on our manuscript (ID: antibiotics-1297681) entitled: “N-Acetylcysteine protects bladder epithelial cells from bacterial invasion and displays antibiofilm activity against urinary tract bacterial pathogens” which have helped us to strengthen the manuscript. Below we present point-by-point answers to all the reviewers’ comments. We have also uploaded the revised version of the manuscript and figures as well as our responses to reviewers in review forum. All changes in the manuscript are highlighted in yellow. I hope the manuscript will now be acceptable to you and the reviewers for publication in “Antibiotics”.
Yours Sincerely
Dr. Theerthankar Das, on behalf of all authors.
Reviewer 3
Page 2, lines 49-54.
“Uropathogenic Escherichia coli (UPEC) is the most prevalent UTI-causing bacterial species, accounting for over 90% of diagnosed community acquired and nosocomial UTIs6. These mainly arise from the colonisation of the periurethral mucosa by faecal bacteria that ascend through the urinary tract7,8. UPEC accounts for 80% and 33% of all uncomplicated and complicated infections, respectively9, 10. Other uropathogens such as Enterococcus spp. account for around 15% of all complicated UTIs in the hospital setting11.
Please analyze the quoted data. E. coli accounts for approximately 90% of community acquired UTIs, but the rates are much lower for hospital acquired UTIs.
Answers to reviewers: Thank you for pointing this out. We have now amended the quoted data to more accurately represent the percentage of hospital acquired UTIs caused by E. coli. and added new reference:
“Uropathogenic Escherichia coli (UPEC) is the most prevalent UTI-causing bacterial species, accounting for over 90% of diagnosed community acquired and 50% of nosocomial UTIs respectively6,7 (refer line 50-51 and reference # 7).
-------------------------------------
Figure 1
Why did the Authors use a line chart to illustrate the obtained results?
Answers to reviewers: Thank you for your question. We thought a line chart aptly illustrates the gradual concentration dependent decrease in live planktonic bacteria present in Figure 1 A and B, and similarly with combinations of NAC in Figure 1 C and D.
------------------------------------------
In the Materials and Methods section, the Authors describe that bacterial cultures were incubated in the presence of 4.57, 4.81, 8.16 g / L of NAC (L 394), while the graph shows (Fig. 1A, 1B) that these values were different. Please explain.
Answers to reviewers: Thank you for pointing this out. We investigated a range of concentrations including the above stated concentrations. We have now amended this to state all tested concentrations. (Current line 400-401).
“Bacterial cultures were inoculated (t =0 hr) at an OD600 = 0.1±0.02, into 96-well, flat-bottomed plates (Corning Corp., USA) in the presence of 3.26, 4.89,5.71, 6.53, 8.16 g/L of NAC at t =0.” Changes highlighted in yellow.
Paragraph 2.1.
Imprecise interpretation of the results, e.g. greater than 4.5 g / L (i.e. 100g / L too?),> 80% inhibition,> 50% inhibition (80% is also more than 50%)
Answers to reviewers: We agree the interpretation was too general. We have now modified the interpretation of concentrations. All changes highlighted in yellow.
(Line 131-132). “Except for one E. coli strain, over 90% inhibition of planktonic growth was observed using NAC concentrations greater than 4.5 g/L and less than 8.5 g/L across all remaining.”
We have also modified the percentage inhibitions to be more precise:
(Line 137-139). “while EC AM-2 showed approximately 50% inhibition in biofilm formation.”
-------------------------------------------------------
Page 3, line 128.
The Authors write that ciprofloxacin was tested at the concentrations of 1x MIC and 1.5xMIC. The graph (Figure 1) shows the values of 30 g/L and 50 g/L (50 g is not 1.5x30). In Table 2, the Authors indicated that the tested strains were sensitive to ciprofloxacin. According to EUCAST, the MIC breakpoint of ciprofloxacin for Enterobacterales is 0.25 mg/L, and for Enterococcus spp. 4 mg/L. How will the Authors explain these discrepancies?
Answers to reviewers: We humbly acknowledge our mistake and have corrected the conversion error throughout the manuscript. We also added in the correct ciprofloxacin concentrations used in this study, which were 0.025 and 0.03 g/L, respectively, well within the EUCAST breakpoints. All changes were highlighted in yellow. I have also modified it to 1xMIC and approximately 1.5x MIC (refer line 137-139).
“When NAC was combined with 1x MIC and ~1.5x MIC ciprofloxacin, a similar decrease in inhibition was observed”
----------------------------------------
Figure 2.
What do the red arrows indicate?
Answers to reviewers: The red arrows are pointing to bacterial aggregation around bladder epithelial cells in the presence or absence of NAC. We have now indicated that in text (line 156-157) and Figure 2 legend (Line168-169). Changes highlighted in yellow.
--------------------------------------------
Paragraph 2.2.
Imprecise interpretation of results: “Six strains when treated with ≥4.89 g/L NAC”,
What does it mean: "there was a > 20 fold decrease from approximately 3700 cells / mm2 to 18 cells / mm2 when exposed to 8.16 g / L" 200 fold, 1000 fold?
Answers to reviewers: We acknowledge our typo error. We have now modified this to the correct fold change of “200-fold” (Current lines 153-155). changes highlighted in yellow.
----------------------------------------------------------------
Minor corrections
1) The names of the microorganisms should be italicized.
Answers to reviewers: Thank you for pointing this out. We have now gone through the manuscript and fixed this.
